# UAV Hyperspectral Data Combined with Machine Learning for Winter Wheat Canopy SPAD Values Estimation

**Qi Wang, Xiaokai Chen** **, Huayi Meng, Huiling Miao, Shiyu Jiang and Qingrui Chang ***

College of Nature Resources and Environment, Northwest A&F University, Yangling, Xianyang 712100, China; wqi@nwafu.edu.cn (Q.W.); xiaokaichen@nwafu.edu.cn (X.C.); menghy@nwafu.edu.cn (H.M.); mhl@nwafu.edu.cn (H.M.); jiangsy@nwafu.edu.cn (S.J.)
* Correspondence: changqr@nwsuaf.edu.cn; Tel.: +86-135-7183-5969

**Abstract:** Chlorophyll is an important indicator for monitoring crop growth and is vital for agricultural management. Therefore, rapid and accurate estimation of chlorophyll content is important for decision support in precision agriculture to accurately monitor the SPAD (Soil and Plant Analyzer Development) values of winter wheat. This study used winter wheat to obtain canopy reflectance based on UAV hyperspectral data and to calculate different vegetation indices and red-edge parameters. The best-performing vegetation indices and red-edge parameters were selected by Pearson correlation analysis and multiple stepwise regression (MSR). SPAD values were estimated using a combination of vegetation indices, vegetation indices and red-edge parameters as model factors, two types of machine learning (ML), a support vector machine (SVM), and a backward propagation neural network (BPNN), and partial least squares regression (PLSR) for four growth stages of winter wheat, and validated using independent samples. The results show that for the same data source, the best vegetation indices or red-edge parameters for estimating SPAD values differed at different growth stages and that combining vegetation indices with red-edge parameters gave better estimates than using only vegetation indices as an input factor for estimating SPAD values. There is no significant difference between PLSR, SVM, and BPNN methods in estimating SPAD values, with better stability of the estimated models using machine learning methods. Different growth stages have a large impact on winter wheat SPAD values estimates, with the accuracy of the four growth stage models increasing in the following order: booting < heading < filling < flowering. This study shows that using a combination of vegetation indices and red-edge parameters can improve SPAD values estimates compared to using vegetation indices alone. In the future, the choice of appropriate factors and methods will need to be considered when constructing models to estimate crop SPAD values.

**Keywords:** UAV hyperspectral; SPAD values; vegetation index; red-edge parameters; machine learning

## 1. Introduction

Chlorophyll is an essential pigment for photosynthesis in plants, which absorbs and transmits light energy and serves as a key indicator of crop growth [1]. Additionally, the content of chlorophyll is closely correlated with the nitrogen and health status of crops [2,3]. Timely monitoring of the crop's chlorophyll levels is crucial for nutrient management in the agricultural field.

Traditional chemical approaches used for chlorophyll content measurement are not only irreversibly invasive toward plant leaves but also operationally intricate and time-consuming to implement [4,5]. Recently, remote sensing technology has experienced rapid growth and has demonstrated efficient, rapid, and noninvasive monitoring capabilities. As a result, it has fostered encouraging research results in the estimation of physical and chemical parameters of vegetation. Nevertheless, ground-based remote sensing still necessitates one-point sampling, which is not only laborious but also carries a temporal lag and has a limited spatial range. These limitations restrict their practical use in the

field [6]. Conversely, satellite remote sensing has partially resolved the issue of spatial scale and can conduct quantitative monitoring at a spatial distance. However, it still exhibits large operational cycles, low spatial accuracy, and is sensitive to weather interference [7]. The advent of UAV remote sensing has largely remedied these issues by offering low-cost, flexible, and straightforward high-resolution image acquisition and the ability to carry out timely monitoring over large land areas [8,9]. Thanks to these advantages, UAV remote sensing is increasingly being used to estimate crops' physicochemical parameters, such as the content of chlorophyll [10,11], yield [7], leaf area index (LAI) [12], and above-ground biomass (AGB) [13].

There are numerous approaches for estimating crops' physical and chemical parameters, with statistical approaches based on vegetation index (VI) being well-liked for their simplicity and resiliency [4,11,14,15]. For instance, Cui et al. [16] compared the efficacy of 12 vegetation indices in estimating the wheat leaf chlorophyll content (LCC) and demonstrated that the red-edge chlorophyll absorption index/triangular vegetation index (RE-CAI/TVI) presented the most superior accuracy out of all the indices. Croft et al. [17] assessed the functionality of 47 vegetation indices in estimating the canopy chlorophyll content of different tree species. They found that the double difference vegetation index (DDVI) exhibited the most robust relation with chlorophyll at the canopy level. Cui and Zhou [18] assessed the sensitivity of various vegetation indices to herbaceous chlorophyll content. The transformed chlorophyll absorption reflectance index/optimized soil-adjusted vegetation index (TCARI/OSAVI) was identified as one of the most appropriate vegetation indices for the estimation of leaf chlorophyll content (LCC). With increased knowledge of plant spectral properties, the strong correlation between reflectance in the red-edge region (680–760 nm) and the physicochemical properties of crops has received significant attention. Ju et al. [19] analyzed the suitability of red-edge symmetry (RES) for estimating leaf chlorophyll content using UAV-acquired hyperspectral data of oilseed rape and wheat and found it to be highly feasible. Boochs et al. [20] analyzed the variability of red-edge reflectance and showed that red-edge characteristics can be influenced by plant biological parameters. Filella and Penuelas [21] investigated the correlation between red-edge reflectance and crop growth parameters and identified that the red-edge position and shape can serve as indicators of plant chlorophyll content, above-ground biomass, and water status. Analysis of the red-edge area of the crop, where reflectance contains a large amount of growth information, can improve the accuracy of the estimation of physical and chemical parameters.

Establishing an experience model based on the relationship between spectral information and SPAD values has become one of the most popular methods for estimating plant SPAD values. Single variable models constructed by spectral parameters usually only consider a small number of bands, especially when processing hyperspectral data, hence the relationship between spectral data and interesting physical and chemical parameters cannot be accurately captured [22]. PLSR is widely considered to be a powerful alternative to a single variable model, and it performs better in most cases [23–25]. In recent years, machine learning has been widely used for estimating crop parameters [26,27]. Liu et al. [28] and Shi et al. [29] have explored the potential performance of machine learning methods, such as SVM, ANN, and BPNN. Machine learning methods have the advantage of avoiding the exploration of crop physiological processes while allowing for the rapid integration of multiple sources of data for non-linear calculations [30]. The studies conducted by Kiala et al. [31] and Yuan et al. [32] indicate that utilizing machine learning algorithms is more effective than the PLSR method. Meanwhile, Kiala et al. [31] and Yuan et al. [32] have indicated that the performance of both PLSR and machine learning methods is influenced by the growth stages of the crops. However, the impact of different methods and growth stages on the model accuracy for estimating SPAD in the canopy of winter wheat using machine learning methods has not been thoroughly investigated.

This study utilized UAV hyperspectral data to screen the optimum vegetation indices and red-edge parameters via a Pearson correlation analysis and multiple stepwise regres-

sion (MSR) methodologies. The combination of VIs and red-edge parameters was then used as model factors. Two machine learning methods, support vector machine (SVM) and backward propagation neural network (BPNN), as well as partial least squares regression (PLSR), were used to estimate the SPAD values of winter wheat. The study aimed to: (1) investigate the impact of VIs and red-edge parameters on the model algorithm; (2) evaluate the potential of SPAD values estimation at different stages of growth; and (3) compare the validity of SPAD values estimation in winter wheat under different methods. This study provides new ideas and methods for rapid, non-destructive, and real-time chlorophyll monitoring using UAVs.

## 2. Materials and Methods

### 2.1. Experimental Profile

The study site is situated in Liangshan Town, Qian County, Xianyang City, Shaanxi Province, China, at coordinates 108°07′E and 34°38′N (Figure 1). It is in the transition zone between the southern region of the Loess Plateau and the Guanzhong Plain, and the landform primarily comprises gullies and hills. The study area has an average altitude of 831 m, and the soil type is red soil. The soil is heavy and lumpy, with little organic matter and a lack of alkali metals. The climate is a warm temperate semi-humid continental monsoon climate, characterized by an average annual temperature of 13.1 °C and an average annual precipitation of 630 mm. The crop maturity period is annual, and the main food crops are summer corn and winter wheat.

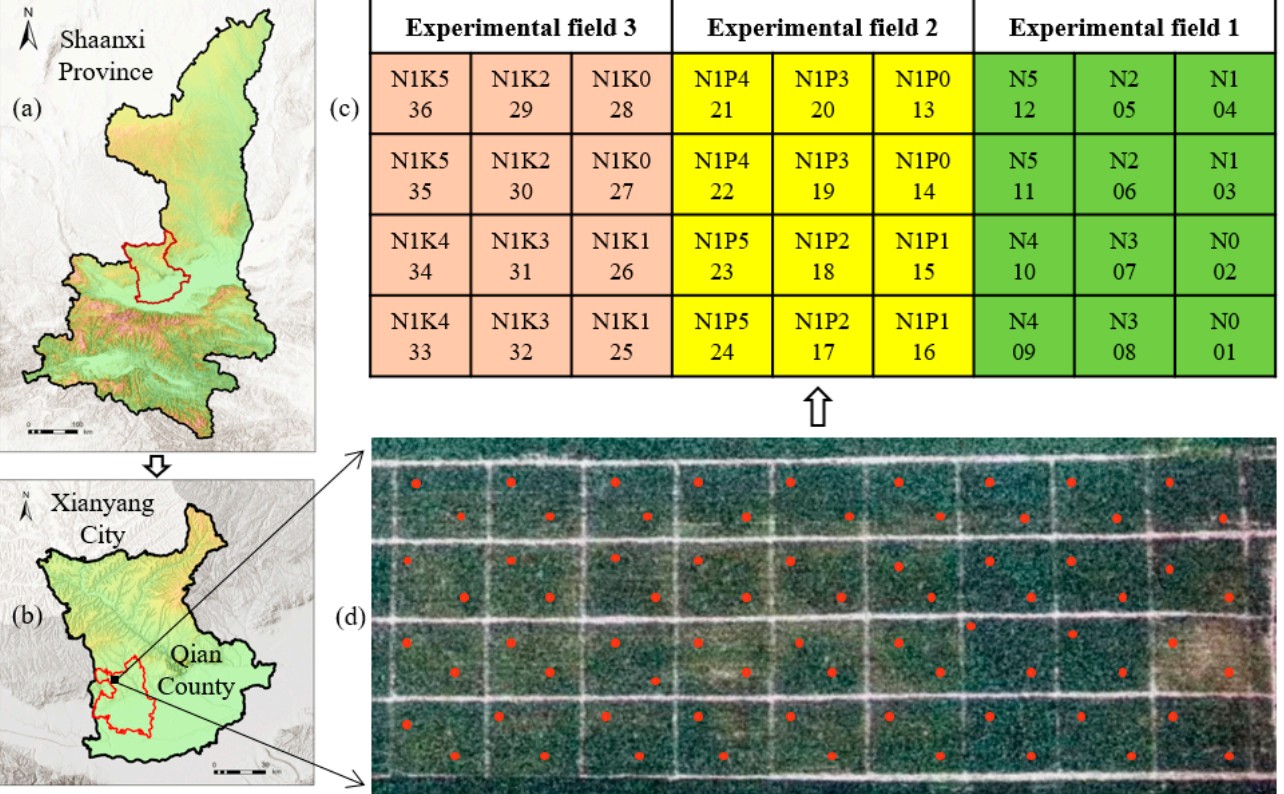

**Figure 1.** (**a**) Xianyang City is located in Shaanxi Province. (**b**) The trial area is located in Xianyang City. (**c**) The fertilizer design for the trial area. (**d**) The distribution of test plots and sampling points.

Winter wheat in the test area was planted on 1 October 2021, and harvested on 12 June of the following year, and the test variety was Xiaoyan 22. A total of 36 test plots were set up for the trial, with each plot measuring $10 \times 9$ m$^2$. To create growth variations between plots, three fertilizers with different nitrogen (N), phosphorus (P$_2$O$_5$), and potassium (K$_2$O) levels were applied (Figure 1). Six levels of each fertilizer were set, with a total of 18 treatments,

each replicated twice. The nitrogen levels were 0, 60, 120, 180, 240, and 300 kg ha$^{-1}$, the phosphate levels were 0, 30, 60, 90, 120, and 150 kg ha$^{-1}$, and the potash levels were 0, 30, 60, 90, 120, and 150 kg ha$^{-1}$. For the observations, two sampling points were set up in each experimental plot, for a total of 72. For four crucial growth stages of winter wheat, CCC and UAV hyperspectral data were obtained. The specific information on the acquired data is shown in Table 1.

**Table 1.** Specific information for each experiment.

| Growth Stage | Date of Measurement | Number of Samples |
|---|---|---|
| Booting | 10 April 2022 | 72 |
| Heading | 25 April 2022 | 72 |
| Flowering | 7 May 2022 | 72 |
| Filling | 23 May 2022 | 72 |

*2.2. Data Collection*

2.2.1. Acquisition and Processing of UAV Hyperspectral Data

Hyperspectral images of winter wheat were acquired using a six-rotor UAV (DIJ M600 Pro) with a Cubert UHD185 (UHD185) imaging hyperspectrometer, as shown in Figure 2. The UAV has a take-off weight of 15.5 kg and a net payload of 9.5 kg, the maximum communication distance is 5 km, and the single battery lasts approximately 30 min, with a pre-set route before the flight and an autonomous return at the end of the flight. The UHD185 camera has a 470 g weight, a 450 nm to 950 nm wavelength range, an 8 nm spectral resolution, and a 4 nm sampling interval. UAV hyperspectral data acquisition is selected when the weather is clear, windless, and cloudless, with a solar altitude angle > 45°, between 10:30 and 14:00. Before each flight, a reference plate is used to perform radiation correction on the spectrometer, which means collecting the reflectance of the spectral correction panel before each flight. This radiation correction is automatically completed within UHD185 and does not require manual processing in the software. When the drone is flying, the lens is vertically downwards with a focal length of 25 mm, a set altitude flight height of 100 m, a speed of 6 m s$^{-1}$, a setting of 80% collateral overlap and 60% heading overlap, and markers at ground sampling points.

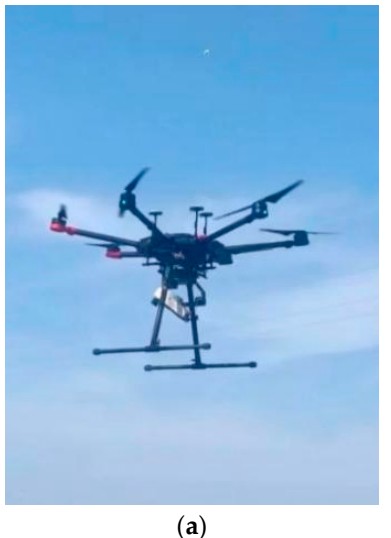 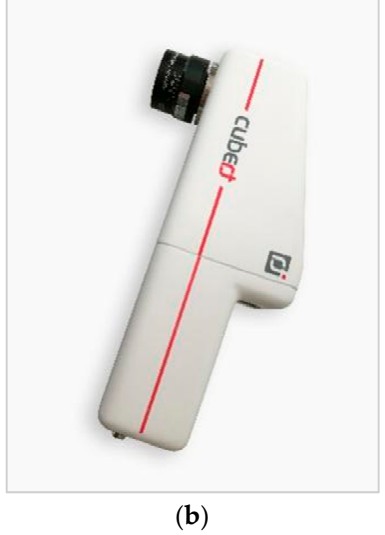 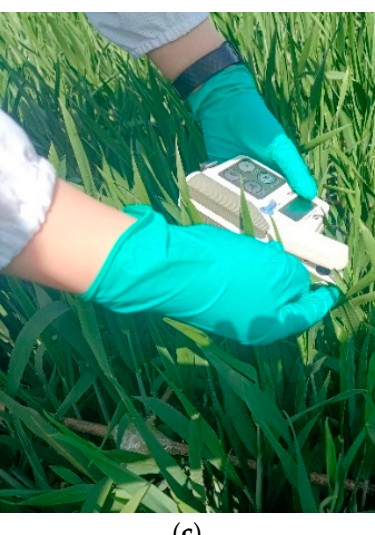

(a)          (b)          (c)

**Figure 2.** (**a**) UAV in working condition. (**b**) UHD185 imaging spectrometer. (**c**) SPAD values measurement.

Individual grey-scale images are fused and stitched together after acquisition with the aid of Cubert Cube Pilot 1.4 and Agisoft PhotoScan Professional 1.16 software. The stitched grey images were then geo-aligned in ArcMap 10.6 and Google Earth 7.3.2 software [33].

The reflectance of the region of interest (ROI) was then constructed and extracted from the ground markers using ENVI 5.1 based on the markers placed at the ground testing points, and the average reflectance of the images within the ROI range was used as the spectral reflectance of the winter wheat canopy at the corresponding sample points [34,35].

### 2.2.2. Measurement of Canopy SPAD Values

The SPAD-502 (Soil and Plant Analyzer Development) portable chlorophyll meter, created in Japan, was used in this investigation to measure the relative chlorophyll content of winter wheat (Figure 2). The SPAD-502 portable chlorophyll meter measuring SPAD values indicating chlorophyll content has been validated: with a correlation coefficient of up to 0.99, plant chlorophyll content and SPAD values exhibit an extremely significant positive association, and the instrument measures the transmission of radiation through leaves at infrared (R) and near-infrared (NIR) wavelengths [6,36,37]. The reading of the SPAD chlorophyll meter, which gauges the relative chlorophyll content of leaves, yields the SPAD value, a dimensionless quantity [38]. The SPAD chlorophyll meter's key benefit is its ability to measure leaf chlorophyll content, damage-free, in situ in real-time, which makes it ideal as a counterpart to drone measurements.

Winter wheat SPAD values after completion of each drone data acquisition were measured. In an area of $0.3 \times 0.3$ m$^2$ area, the top leaves of different plants were selected for the determination, and a total of 20 leaves with uniform growth were selected, and an average of three measurements were taken at various points on each leaf to determine its SPAD values. The SPAD values of the sample size was determined by taking the average of the 20 leaves. The region is used as the ROI to extract the UAV hyperspectral reflectance. Figure 1 and Table 1 display the sampling locations and sample count.

### 2.3. Vegetation Indices and Red-Edge Parameters

The vegetation index was obtained by linear or non-linear combinations of spectral reflectance in different bands, and green plants have distinctive spectral characteristics in the red-edge band range from other landscape features. Based on this, this study builds on previous studies to construct the vegetation indices and red-edge parameters shown in Table 2.

**Table 2.** Formulas or definitions of vegetation indices (VIs) or red-edge parameters.

| Vegetation Index or Red-Edge Parameter | Formula or Definition | Reference |
|---|---|---|
| ratio vegetation index (RVI) | R800/R760 | [39] |
| green normalized difference vegetation index (GNDVI) | (R780 − R550)/(R780 + R550) | [40] |
| plant biomass index (PBI) | R810/R560 | [41] |
| leaf chlorophyll index (LCI) | (R850 − R710)/(R850 + R670)$^{1/2}$ | [42] |
| simple ratio index (SR) | R750/R550 | [43] |
| optimize soil and adjust vegetation index (OSAVI) | $1.16 \times$ (R800 − R670)/(R800 + R670 + 0.16) | [44] |
| structure insensitive pigment index (SIPI) | (R810 − R460)/(R810 + R460) | [45] |
| pigment ratio vegetation index (PRVI) | R800/R553 | [46] |
| photochemical reflectance index (PRI) | (R570 − R531)/(R570 + R531) | [47] |
| red-edge chlorophyll index (CI$_{red-edge}$) | R800/R720 − 1 | [48] |
| Dr | the maximum value of the first derivative the spectrum of the red-edge region | [7] |
| Dr$_{min}$ | minimum red-edge amplitude | [7] |
| Dr/Dr$_{min}$ | red-edge amplitude/minimum amplitude value | [7] |
| SDr | the sum of the first-order differential of the red-edge region spectrum | [21] |
| RES | the ratio of 718 nm left red-edge area to the whole red-edge area | [19] |

Note: Ri denotes the reflectance at wavelength i. If there is no corresponding Ri, the average of the reflectance at adjacent wavelengths is used.

*2.4. Selection of Vegetation Indices and Red-Edge Parameters and Regression Analysis Methods*

2.4.1. Selection of Vegetation Indices and Red-Edge Parameters

The absolute value of the kurtosis of the variables in this study is less than 10, the absolute value of the skewness is less than 3, and the data are basically characterized by normal distribution. The Pearson correlation analysis was used to study the correlation between VIs or red-edge parameters and SPAD values in wheat, implemented in SPSS PRO 1.0.11 software. The equation is shown in Equation (1):

$$r = \frac{\sum_{i=1}^{n}(x_i - \overline{x})(y_i - \overline{y})}{\sqrt{\sum_{i=1}^{n}(x_i - \overline{x})^2}\sqrt{\sum_{i=1}^{n}(y_i - \overline{y})^2}}, \tag{1}$$

where $x_i$ and $y_i$ represent the sample values, $\overline{x}$ and $\overline{y}$ are the mean values of the sample set, $n$ is the number of samples, $r$ represents the correlation coefficient, when $|r|$ is larger, the higher the correlation between $x$ and $y$.

Multiple stepwise regression (MSR) is an improvement on multiple linear regression (MLR) in terms of efficiency, a statistical method for selecting the optimal variable from multiple independent variables [49]. Following the determination of the first set of variables, the variable outside the set with the largest impact on the dependent variable is chosen, and each time a brand-new independent factor is introduced, it is compared with the existing variables in the set. The variables that are correlated and those that have the least influence on the dependent variable are removed, in turn, until the number of variables does not increase [50]. The MSR for this study was implemented in SPSS PRO 1.0.11 software.

Pearson correlation analysis was used to select VIs and red-edge parameters that were highly significantly ($p < 0.01$) correlated with SPAD values. The optimal VIs and red-edge parameters for each growth stage were then selected using MSR. Finally, the intersection of the two was chosen to build a model for estimating SPAD values.

2.4.2. Regression Analysis Method

Using the statistical analysis approach known as partial least squares regression (PLSR), principal component analysis (PCA), conventional correlation analysis (CCA), and multiple linear regression (MLR) are all carried out simultaneously [51]. For hyperspectral data, PLSR can both achieve data dimensionality reduction and be used as a modeling method to solve the covariance problem of the independent variables while maximizing the extraction of spectral feature information, effectively enhancing the adaptability of the model [7]. This study implemented PLSR in The Unscrambler X 10.4 software.

Support vector regression (SVM) is a statistically based implementation of structural risk minimization approximation. SVM can adapt to regression problems with high-dimensional features. At its core, it creates an ideal place in the high-dimensional space to finish the classification process and then utilizes inverse mapping to return to the low-dimensional space thereafter. The method employs a kernel function to nonlinearly transfer data from a low-dimensional space to a high-dimensional space [52]. The dimensionality of the input data has no bearing on how complicated the computation outputs are, effectively avoiding over-fitting problems and having a strong generalization capability [53]. To obtain better estimation results, this article uses a grid search method to determine the penalty factor (c) and kernel function parameter (g). The optimal values of c and g are selected within the range of [$10^{-2}$, $10^{-1}$, 1, 10, 100] and [$10^{-4}$, $10^{-3}$, $10^{-2}$, $10^{-1}$, 1, 10], respectively. This study implemented the SVM method in Matlab 2019b software.

A multilayer feedforward neural network called a backward propagation neural network (BPNN) propagates backward by the error. The BPNN is a popular tool for estimating the physical and chemical properties of crops because it is efficient at resolving non-linear multi-dimensional fitting issues in complicated regressions [54]. The BPNN is made up of three layers: an input layer, an implicit layer, and an output layer. The BPNN can have multiple implicit layers, and the precise number of layers will need to be determined after extensive testing. Neurons within the same implicit layer are not

connected, but they are fully connected to those in adjacent implicit layers [53]. The two processes that make up BPNN are forward signal propagation and backward error propagation, where the error output is determined in the forward direction, and the weights and thresholds are changed from the reverse way. The input and output layer neuron parameters of BPNN are determined by the number of independent and dependent variables, and the number of nodes in the hidden layer is calculated using an empirical formula. The BPNN method for this study was implemented in Matlab 2019b software.

### 2.5. Modeling Set and Verification Set Division

The standard deviation method was used to identify outliers for SPAD values measurements at each growth stage, i.e., values measured outside the ($\mu - 3\sigma$, $\mu + 3\sigma$) range in a set of data were considered outliers, and 286 samples remained after the removal of 2 outliers. Then, at each growth stage, the values were arranged in order of ascending SPAD values, using a stratified sampling ratio of 2:1. The number of samples obtained for the modeling set was 47, 47, 48, and 48 for the booting, heading, flowering, and filling stages, and the number of samples for the validation set was 24 in all cases. SPAD values estimation models for each growth stage were developed using the modeling set data and model testing was carried out using the corresponding validation set samples.

### 2.6. Model Accuracy Testing

The accuracy of the SPAD values estimating model was examined in this research using the coefficient of determination ($R^2$), root mean square error (RMSE), and relative percentage difference (RPD). In other words, the more closely $R^2$ approaches 1, the lower the RMSE, the greater the RPD, and the more accurate the model. When RPD is less than 1.4, the model is unable to predict the sample; between 1.4 and 2, the model has a rough capacity to forecast; and when RPD is more than 2, the model has an incredibly powerful ability to predict [10].

$$R^2 = \frac{\sum_{i=1}^{n}(\hat{y}_i - \bar{y})^2}{\sum_{i=1}^{n}(y_i - \bar{y})^2},\tag{2}$$

$$RMSE = \sqrt{\frac{1}{n}\sum_{i=1}^{n}(\hat{y}_i - y_i)^2},\tag{3}$$

$$RPD = \frac{SD}{RMSE}\sqrt{\frac{n}{n-1}},\tag{4}$$

where $\hat{y}_i$ and $y_i$ represent the predicted and measured values of the samples, $\bar{y}$ is the mean of the measured samples, *n* is the number of samples, and *SD* is the variance of the measured values of the samples.

## 3. Results

### 3.1. Descriptive Statistical Analysis of Canopy SPAD Values

Table 3 displays SPAD values data for the four development phases of winter wheat. The statistical characteristics demonstrated statistical similarities between the modeling and validation sets, with the winter wheat SPAD values varying between 10.53 and 62.00 and the coefficient of variation (CV) ranging from 0.06 to 0.27. As the wheat developed, the mean SPAD values displayed a pattern of growing and then declining, with the greatest occurring during flowering and the lowest occurring at filling, the maximum and minimum values of SPAD values occurred at filling, the stage with the largest span of SPAD values. At filling, the maximum standard deviation (SD) and coefficient of variation (CV) values also occurred.

**Table 3.** Statistical characteristics of canopy SPAD values at different growth stages.

| Data Sets | Growth Stages | Number of Samples | MIN | MEAN | MAX | SD | CV (%) |
|---|---|---|---|---|---|---|---|
| Modeling set | Booting | 47 | 40.2 | 49.37 | 56.83 | 3.88 | 0.08 |
| | Heading | 47 | 42.07 | 49.68 | 55.40 | 3.09 | 0.06 |
| | Flowering | 48 | 34.90 | 50.20 | 59.10 | 5.62 | 0.11 |
| | Filling | 48 | 10.53 | 46.01 | 62.00 | 12.09 | 0.26 |
| Validation set | Booting | 24 | 38.7 | 49.11 | 55.27 | 4.17 | 0.08 |
| | Heading | 24 | 41.67 | 49.53 | 55.23 | 3.28 | 0.07 |
| | Flowering | 24 | 37.07 | 50.27 | 58.60 | 5.56 | 0.11 |
| | Filling | 24 | 11.97 | 45.91 | 59.90 | 12.33 | 0.27 |

### 3.2. Correlation of Canopy SPAD Values with Vegetation Indices or Red-Edge Parameters

Figure 3 displays the results of the correlation ($|r|$) study between winter wheat SPAD values and VIs or red-edge characteristics based on the Pearson correlation analysis. The results demonstrated that the relationships between SPAD values and Vis or red-edge characteristics were not constant throughout the development stages. During the whole growth stage, $|r|$ increases as the winter wheat grows, with the highest $|r|$ at the filling stage. For Vis, all Vis showed a highly significant correlation ($p < 0.01$) at all four growth stages, except LCL and PRI which did not reach a highly significant correlation at booting ($p < 0.01$). For the red-edge parameters, Dr, $Dr_{min}$, and RES showed a highly significant correlation throughout reproduction ($p < 0.01$), $Dr/Dr_{min}$ showed a highly significant correlation only at the filling stage ($p < 0.01$), and SDr showed a highly significant correlation except at booting ($p < 0.01$).

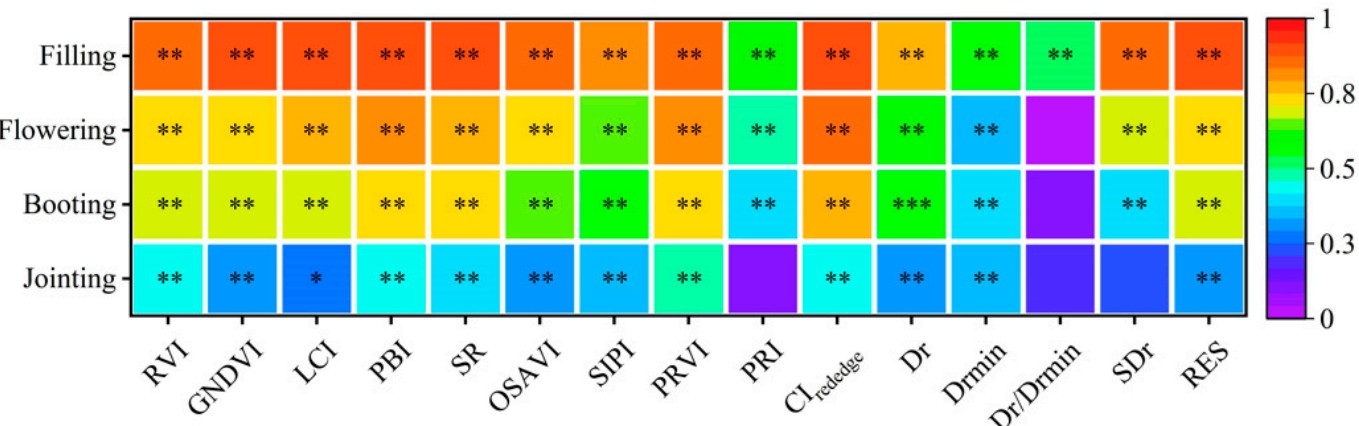

**Figure 3.** Correlation between SPAD values and vegetation indices or red-edge parameters at each stage of growth. Note: * indicates significant at the 0.05 level ($p < 0.05$), ** indicates significant at the 0.01 level ($p < 0.01$), *** indicates significant at the 0.001 level ($p < 0.001$).

At booting, PRVI had the highest correlation coefficient ($|r| = 0.48$) among the VIs and $Dr_{min}$ had the highest correlation ($|r| = 0.36$) among the red-edge parameters. At heading, it had the highest correlation of $CI_{red-edge}$ ($|r| = 0.75$) in VIs and RES ($|r| = 0.69$) in the red-edge parameters. At flowering, the $CI_{red-edge}$ correlation was highest in VIs ($|r| = 0.87$) and the RES correlation was highest in the red-edge parameters ($|r| = 0.72$). At filling, the correlation was highest for LCL in VIs ($|r| = 0.91$) and RES in the red-edge parameters ($|r| = 0.90$).

### 3.3. Based on MSR Estimation of SPAD Values Using Vegetation Indices or Red-Edge Parameters and Variable Selection

The winter wheat SPAD values were estimated using a multiple stepwise regression (MSR) method based on the VIs or red-edge parameters in Table 2 and the results are shown in Table 4 and Figure 4. At booting, based on the VIs, the modeling $R^2$ value was

0.26 (RMSE = 3.31, RPD = 0.63), and based on the red-edge parameters, the modeling $R^2$ value was 0.20 (RMSE = 3.42, RPD = 0.50). For this stage, the best-performing VIs and red-edge parameters were PRVI and $Dr_{min}$. At heading, based on the VIs, the modeling $R^2$ value was 0.59 (RMSE = 1.979, RPD = 1.28), and based on the red-edge parameters, the modeling $R^2$ value was 0.41 (RMSE = 2.34, RPD = 0.87). For this stage, the best performing VIs and red-edge parameters were $CI_{red-edge}$, RVI, and RES. At flowering, based on the VIs, the modeling $R^2$ value was 0.84 (RMSE = 2.21, RPD = 2.30), and based on the red-edge parameters, the modeling $R^2$ value was 0.53 (RMSE = 3.81, RPD = 1.06). For this stage, the best performing VIs and red-edge parameters were $CI_{red-edge}$, RVI, PBI, and RES. At filling, based on the VIs, the modeling $R^2$ value was 0.84 (RMSE = 4.8, RPD = 2.28), and based on the red-edge parameters, the modeling $R^2$ value was 0.86 (RMSE = 4.53, RPD = 2.41). For this stage, the best-performing VIs and red-edge parameters were GNDVI, RES, and SDr.

**Table 4.** SPAD values estimates based on vegetation indices or red-edge parameters and multiple stepwise regression method.

| Growth Stage | Model Factors | Regression Equation | Optimal Vegetation Indices or Red-Edge Parameters |
|---|---|---|---|
| Booting | VIs | $y = 39.80 + 0.96 \times PRVI$ | PRVI |
| | REPs | $y = 51.39 + 23{,}760.34 \times Dr_{min}$ | $Dr_{min}$ |
| Heading | VIs | $y = 35.44 + 8.27 \times CI_{red-edge} - 0.51 \times RVI$ | $CI_{red-edge}$, RVI |
| | REPs | $y = 61.58 - 70.68 \times RES$ | RES |
| Flowering | VIs | $y = 27.68 + 8.35 \times CI_{red-edge} - 2.23 \times RVI + 4.14 \times PBI$ | $CI_{red-edge}$, RVI, PBI |
| | REPs | $y = 79.56 - 139.52 \times RES$ | RES |
| Filling | VIs | $y = -34.71 + 131.62 \times GNDVI$ | GNDVI |
| | REPs | $y = 103.18 - 148.61 \times RES - 10.43 \times SDr$ | RES, SDr |

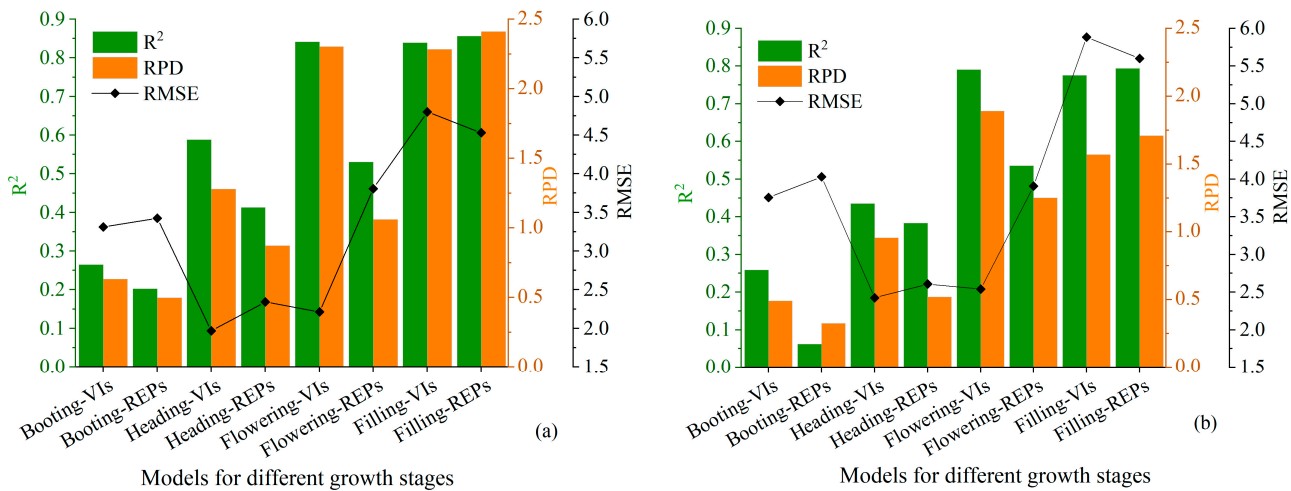

**Figure 4.** Estimation of SPAD values by growth stage based on multiple stepwise regression. (**a**) Modeling set; (**b**) Validation set.

Based on VIs, there is an increase in modeling $R^2$ from 0.26 to 0.84 from the booting to the filling stage. The modeling's smallest RMSE value was at the heading stage (RMSE = 1.97) and the largest at the filling stage (RMSE = 4.80). The modeling's smallest RPD value was at the booting stage (RPD = 0.63) and the largest at flowering (RPD = 2.30). Based on the red-edge parameters, the modeled $R^2$ increases from 0.20 to 0.86 from the booting to the filling stage. Modeling RMSE values were smallest at the heading stage (RMSE = 2.34) and largest at the filling stage (RMSE = 4.53). There is an increase in modeling RPD from 0.50 to 2.41 from the booting stage to the filling stage. In addition, as shown in Figure 4, the results for the validation set $R^2$, RMSE, and RPD remain largely consistent

with the results of the corresponding modeling set. For example, the larger the RPD of the modeling sample, the larger the RPD of the validation sample.

Based on the Pearson correlation analysis, VIs showing highly significant correlations ($p < 0.01$) were RVI, GNDVI, PBI, PRI, SR, OSAVI, SIPI, and PRVI. The red-edge parameters that showed a highly significant correlation ($p < 0.01$) were Dr, $Dr_{min}$, and RES. Based on the MSR method, the best-performing VIs for the four growth stages were PRVI, $CI_{red-edge}$, RVI, PBI, and GNDVI, and the best-performing red-edge parameters were $Dr_{min}$, RES, and SDr. Combining the above two methods to select modeling factors, for the VIs, PRVI, $CI_{red-edge}$, RVI, PBI, and GNDVI were selected to construct the SPAD values estimation model, for the red-edge parameters, $Dr_{min}$ and RES were selected to construct the SPAD values estimation model.

### 3.4. Estimation of Canopy SPAD Values Using PLSR, SVM, and BPNN Methods

To achieve SPAD values estimation in winter wheat at different growth stages, based on the Pearson correlation analysis and MSR methods, five VIs (PRVI, $CI_{red-edge}$, RVI, PBI, GNDVI) and two red-edge parameters ($Dr_{min}$, RES) were selected. The following methodology was used for modeling: (1) three methods of PLSR, SVM, and BPNN regression; (2) five VIs based on hyperspectral UAV data; (3) five VIs combined with two red-edge parameters. The modeling results are shown in Figures 5 and 6.

When using only VIs to estimate winter wheat SPAD values, in all models, for the PLSR, the best modeling $R^2$ value was 0.85 (RMSE = 2.16, RPD = 2.37); for the SVM, the best modeling $R^2$ value was 0.88 (RMSE = 4.2, RPD = 2.65); for the BPNN, the best modeling $R^2$ value was 0.85 (RMSE = 2.15, RPD = 2). At booting, the BPNN regression model was the best with an $R^2$ value of 0.30 (RMSE = 3.23, RPD = 0.72). At heading, the PLSR model was the best with an $R^2$ value of 0.64 (RMSE = 1.81, RPD = 1.36). At flowering, the SVM regression model was the best with an $R^2$ value of 0.85 (RMSE = 2.20, RPD = 2.23). At filling, the SVM regression model was best with an $R^2$ value of 0.89 (RMSE = 4.10, RPD = 2.61).

When using the combination of VIs and red-edge parameters to estimate SPAD values, in all models, for PLSR, the best modeling $R^2$ value was 0.88 (RMSE = 1.90, RPD = 2.77); for SVM, the best modeling $R^2$ value was 0.88 (RMSE = 4.20, RPD = 2.65); for BPNN, the best modeling $R^2$ value was 0.86 (RMSE = 4.50, RPD = 2.47). At booting, the BPNN regression model was the best with an $R^2$ value of 0.46 (RMSE = 2.83, RPD = 0.89). At heading, the PLSR model was the best with an $R^2$ value of 0.69 (RMSE = 1.71, RPD = 1.45). At flowering, the PLSR model was best with an $R^2$ value of 0.88 (RMSE = 1.9, RPD = 2.77). At filling, the SVM regression model was best with an $R^2$ value of 0.88 (RMSE = 4.20, RPD = 2.65).

In addition, the results of the validation set are largely consistent with the results of the modeling set, as shown in Figure 6. That is, the higher the modeling $R^2$ and RPD, the larger the validation $R^2$, and the higher the RPD, the lower the modeling RMSE, the smaller the validation RMSE. The results also indicate that there are no significant differences between the PLSR, SVM, and BPNN methods in estimating SPAD values. The optimum growth stage for SPAD values estimation was blooming, and utilizing a combination of red-edge parameters and VIs to estimate SPAD values was more accurate than using VIs alone. Different growth stages had a significant influence on the accuracy of the estimation model.

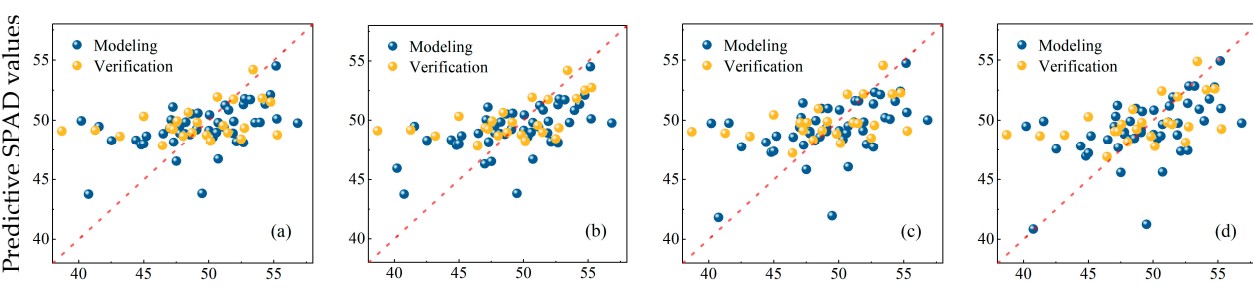

**Figure 5.** *Cont.*

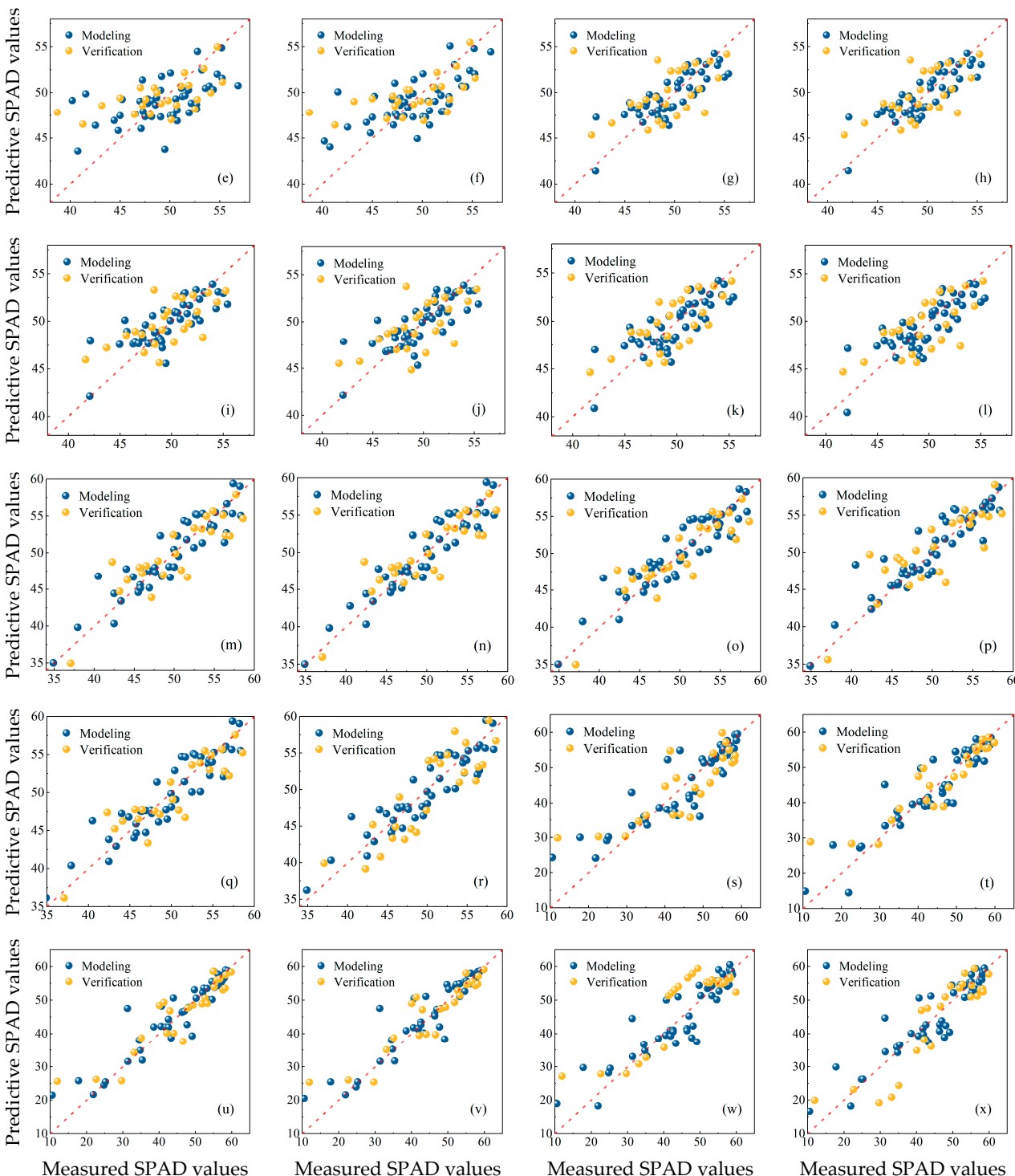

**Figure 5.** The prediction results of canopy SPAD values were estimated using PLRS, SVM, and BPNN methods for each growth stage. (**a**) Booting, using VIs based on PLSR; (**b**) Booting, using VIs and REPs based on PLSR; (**c**) Booting, using VIs based on SVM; (**d**) Booting, using VIs and REPs based on SVM; (**e**) Booting, using VIs based on BPNN; (**f**) Booting, using VIs and REPs based on BPNN; (**g**) Heading, using VIs based on PLSR; (**h**) Heading, using VIs and REPs based on PLSR; (**i**) Heading, using VIs based on SVM; (**j**) Heading, using VIs and REPs based on SVM; (**k**) Heading, using VIs based on BPNN; (**l**) Heading, using VIs and REPs based on BPNN; (**m**) Flowering, using VIs based on PLSR; (**n**) Flowering, using VIs and REPs based on PLSR; (**o**) Flowering, using VIs based on SVM; (**p**) Flowering, using VIs and REPs based on SVM; (**q**) Flowering, using VIs based on BPNN;

(**r**) Flowering, using VIs and REPs based on BPNN; (**s**) Filling, using VIs based on PLSR; (**t**) Filling, using VIs and REPs based on PLSR; (**u**) Filling, using VIs based on SVM; (**v**) Filling, using VIs and REPs based on SVM; (**w**) Filling, using VIs based on BPNN; (**x**) Filling, using VIs and REPs based on BPNN.

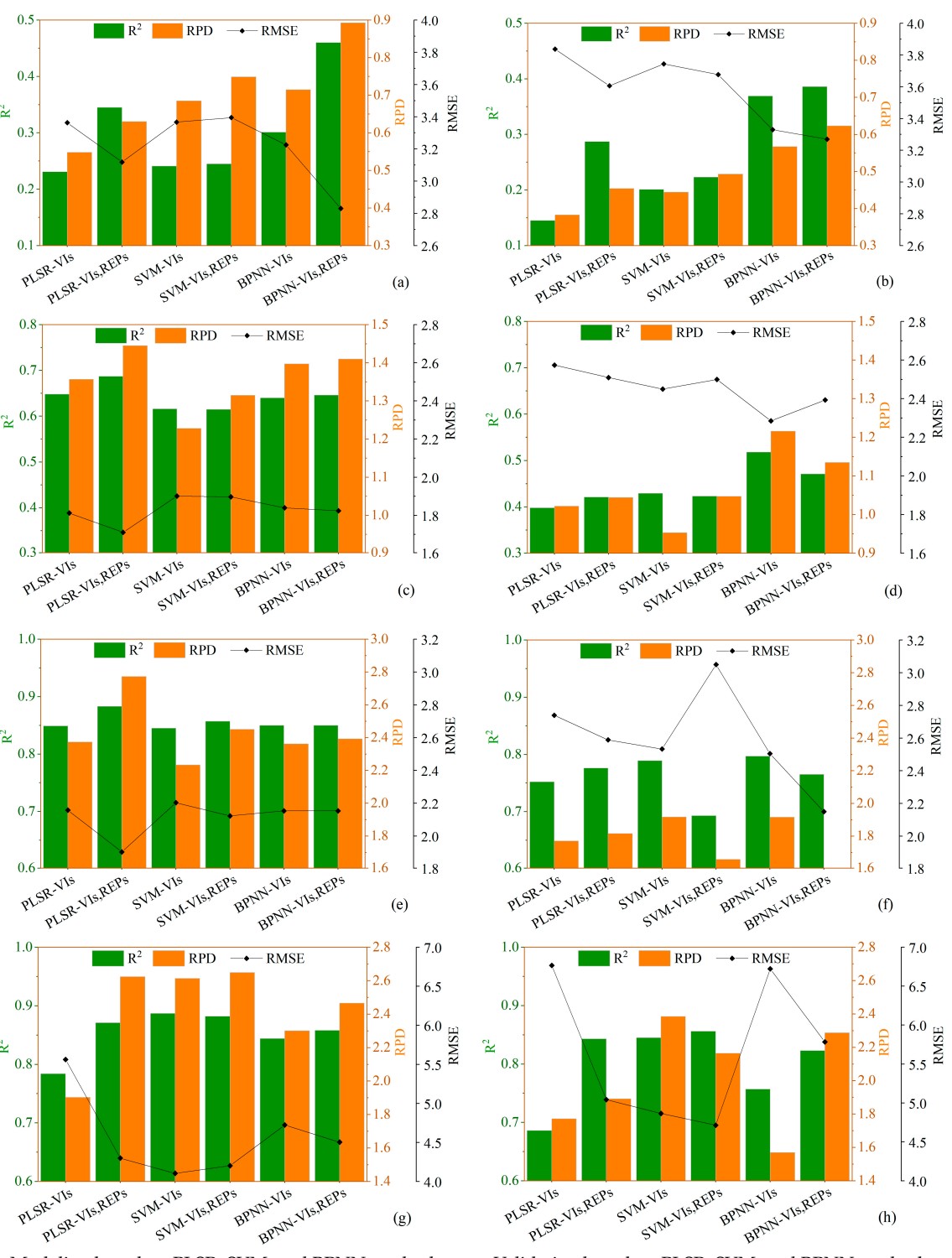

Modeling based on PLSR, SVM, and BPNN methods.    Validation based on PLSR, SVM, and BPNN methods.

**Figure 6.** Accuracy parameters for the modeling and validation set for canopy SPAD Values estimation using PLRS, SVM, and BPNN methods. (**a**) Booting modeling set; (**b**) Booting validation set; (**c**) Heading modeling set; (**d**) Heading validation set; (**e**) Flowering modeling set; (**f**) Flowering validation set; (**g**) Filling modeling set; (**h**) Filling validation set.

## 4. Discussion

### 4.1. Estimation of SPAD Values Using Vegetation Indices, Red-Edge Parameters, and Their Combinations

The research showed that the SPAD values estimate model produced by VIs was better than that produced by the red-edge parameters (Figure 4). According to Gao et al. [55], using only VIs rather than red-edge parameters to evaluate the physiological and biochemical characteristics of crops produced superior results. This result is in line with the conclusion in this research that utilizing VIs alone produced better SPAD values estimations than using the red-edge parameters alone. Compared to the red-edge parameters, the composition of the vegetation index includes visible and near-infrared wavelengths [41]. This indicates that for UHD185, both visible and near-infrared wavelengths can play an important role in estimating chlorophyll content in winter wheat [56].

Based on PLSR, SVM, and BPNN using VIs, VIs were combined with red-edge parameters as model factors to estimate SPAD values. When VIs and red-edge parameters are used together, the accuracy of the CCC estimate is improved compared to when VIs are used alone (Figures 5 and 6). Accordingly, it is possible to increase the precision of the SPAD values estimations for wheat by including red-edge characteristics in the modeling elements. This is because the red-edge is located in the region where the spectral reflectance sharply increases between the red absorption valley and the near-infrared reflection peak, with a general wavelength range of 680–760 nm, which can reflect chlorophyll content and leaf structure. The position of the red-edge is sensitive to the chlorophyll content in the plant canopy and plays an important role in estimating the chlorophyll concentration of vegetation [57,58]. The red-edge parameters, as shown by Tao et al. [59] enhanced the estimations of the leaf area index (LAI) and above-ground biomass (AGB). AGB and LAI are strongly associated with SPAD values, and the results of this work are compatible with their results.

### 4.2. Estimation of Canopy SPAD Values Using PLSR, SVM, and BPNN

A comparison of PLSR, SVM, and BPNN modeling results showed no significant differences between PLSR and machine learning (ML) methods in estimating CCC, which is consistent with the results of Almeida et al. [60] and Zhu et al. [61]. Of the three modeling methods, the PLSR method has poor modeling stability and the BPNN method performs best in SPAD values estimation, which is influenced by the characteristics of the algorithm. When using both SVM and BPNN machine learning methods for modeling, we discovered that the BPNN model was more reliable than the SVM model, which is consistent with the results of Wang et al. [62]. The BPNN method allows for rapid modification of the algorithm and performs better when the sample size is large. The SVM approach has been popular in recent years in remote sensing research because it is appropriate for multidimensional datasets and small samples [63]. In this study, two conventional ML techniques were chosen, and future research should focus on some more advanced prediction methods such as deep learning [64], cubist (CB) method [60], and integration algorithms [65].

In addition, the PLSR, SVM, and BPNN used in this study are only applicable to the region being calibrated and have limitations [66]. Radiative transfer models (RTMs) such as PROSAIL [67], ACRM [68], and SCOPE [69] models have been applied to SPAD values estimation. However, there are certain problems with RTMs, for example, the impact of row crops on model inversion and the impact of ill-posed inverse problems [70,71]. However, different angles of observation may also increase the uncertainty of the model, which is discussed in depth by Duan et al. [72] and Wang et al. [73]. In addition, RTMs and ML were used by Xu et al. [74] to improve the precision of crop chlorophyll content prediction. In general, both simplicity and applicability should be considered when estimating crop agronomic parameters.

Hyperspectral data can provide rich and detailed spectral information and can be better applied to quantitative crop chlorophyll analysis [15,75]. However, hyperspectral data are complex to process in use and sensitive to noise and interference. Some existing studies

have shown that UAV-based multispectral data have good performance in estimating crop canopy chlorophyll [9,76]. Future studies could compare the performance of multispectral and hyperspectral data in estimating chlorophyll in winter wheat.

### 4.3. Canopy SPAD Values Estimates for Winter Wheat at Different Growth Stages

In this study, SPAD values estimation models were constructed for four key growth stages of winter wheat, and the maximum accuracy of the SPAD values estimation models varied at different growth stages. The estimation models that achieve the highest accuracy are BPNN, PLRS, PLSR, and SVM for the booting, heading, flowering, and filling stages, respectively. With the same modeling approach and parameter types, the highest accuracy and stability of SPAD values estimation were achieved for the winter wheat flowering stage. Although the $R^2$ values of some of the models during the filling stage are large, the RMSE is also large, and the stability of the models is poor. The accuracy of the SPAD values estimates increased in the following order: booting < heading < filling < flowering, suggesting that different growth stages do have an effect on SPAD values estimation in winter wheat, which is consistent with the results of Zhu et al. [6]. Each growth stage of winter wheat has different growth characteristics, and four key growth stages of winter wheat were selected for modeling in this study, and further validation is needed to see if these models can be applied to other growth stages.

The accuracy and robustness of SPAD values estimates can be influenced by factors, such as plant structure, leaf thickness, plant cover, and soil context, and these effects vary by growth stage [77]. At the booting and heading, due to the small size of the plants, the soil background has a greater influence on canopy reflectance. At flowering and filling, vegetation cover and leaf area increase, masking the effects of the soil background and reducing visible light and red-edge reflectance [78]. The average spectral reflectance of the four growth stages is shown in Figure 7. In addition, SPAD values vary with increasing leaf size and number, thus having an impact on crop reflectance and SPAD values estimates [77]. As can be seen from Table 3, the highest values of the coefficient of variation (CV) for SPAD values were found during the filling stage and significant SPAD values variation may have contributed to a good fit of the prediction model, but at the same time, the wheat ears and some senescing leaves may have affected the spectral reflectance and affected the accuracy of the model during this growth stage. In addition, previous studies have shown that saturation of VI occurs when the vegetation canopy cover is high and may occur to some extent during the filling stage [79]. To elaborate on these results, future studies should make more frequent observations to collect observations throughout the growth stage of the crop.

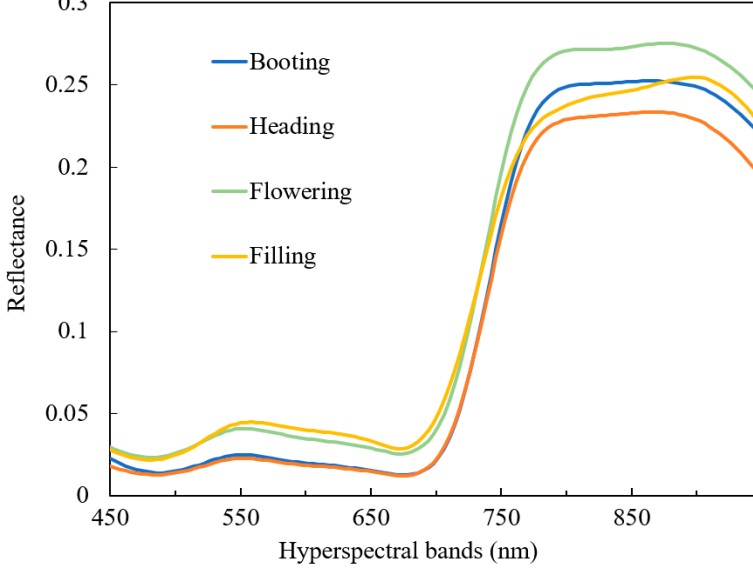

**Figure 7.** Average spectral reflectance of wheat at four growth stages.

## 5. Conclusions

This study estimated the SPAD values of winter wheat at different growth stages using vegetation indices, vegetation indices combined with red edge parameters, PLSR, SVM, and BPNN based on UAV hyperspectral (UHD185) data. The main conclusions are as follows:

(1) Better SPAD values estimates were obtained when vegetation indices alone were used compared to when the red-edge parameters were used alone. The accuracy of the model for estimating SPAD values in winter wheat was better when vegetation indices and red-edge parameters were combined compared to the use of vegetation indices or red-edge parameters.

(2) Using a combination of vegetation indices and red-edge parameters, the predictive performance of PLSR, SVM, and BPNN methods can be improved, with BPNN being better than PLSR and SVM in terms of predictive power and stability.

(3) Different growth stages greatly impacted winter wheat SPAD values estimation, with flowering being the best stage for estimating winter wheat SPAD values. The best-performing model was based on the combination of vegetation indices and red-edge parameters BPNN ($R^2$ = 0.85, RMSE = 2.15, RPD = 2.39).

**Author Contributions:** Conceptualization, Q.W., X.C. and Q.C.; methodology, Q.W., S.J., X.C. and H.M. (Huayi Meng); software, Q.W. and H.M. (Huiling Miao); validation, Q.W., X.C., S.J. and Q.C.; formal analysis, Q.W. and X.C.; investigation, Q.W., S.J., X.C., H.M. (Huiling Miao) and H.M. (Huayi Meng); resources, Q.C. and H.M. (Huayi Meng); data curation, Q.W., S.J., X.C., H.M. (Huayi Meng) and H.M. (Huiling Miao); writing—original draft preparation, Q.W.; writing—review and editing, Q.W., X.C. and Q.C.; visualization, H.M. (Huiling Miao) and X.C.; supervision, Q.C.; project administration, Q.C.; funding acquisition, Q.C. All authors have read and agreed to the published version of the manuscript.

**Funding:** This study was supported by the National High-Tech Research and Development Program (863 Program) of China under Grant No. 2013AA102401-2.

**Data Availability Statement:** Data sharing does not apply to this paper.

**Acknowledgments:** We thank all the students and teachers of Chang's team at Northwest Agriculture and Forestry University for their support in the acquisition of experiments and technical aspects of this study.

**Conflicts of Interest:** The authors declare no conflict of interest.

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
