# Peer review of "UAV Hyperspectral Data Combined with Machine Learning for Winter Wheat Canopy SPAD Values Estimation"

_remotesensing, doi:10.3390/rs15194658_

Round 1

Reviewer 1 Report

It is recommended to improve the English writing.

Reviewer 2 Report

Comments and suggestions:

In this manuscript, the authors combined UAV hyperspectral reflectance data and field measured data for chlorophyll retrieval using different Nonlinear fitting methods, such as PLSR and SVM. Abundant work has been involved in the manuscript, and extensive detailed information on data analysis have been presented. Nevertheless, novelty of the research seemed to be not enough, since the manuscript just used some algorithms to fit reflectance with SPAD values for regression. This is quite similar to some published papers that showed good results for nondestructively estimation pigment contents. The authors should explain clearly the key question that needed to be solved for chlorophyll estimation, and the benefit to use UAV hyperspectral data for chlorophyll estimation improvement, and the difference from other similar published papers. The authors must rethink these key problems of the manuscript and improve its novelty. Moreover, I believe the manuscript should clarify several crucial issues which are presented in the general comments below, and some minor specific comments also should be addressed for the improvement of the manuscript.

General comments

1. in the first part of the manuscript, i.e., introduction section, most content were concerned about methods for chlorophyll estimation, such as spectral indices and machine learning methods. however, limited content was about key problems proposal of the manuscript. Moreover, in the last paragraph, the authors mentioned with UAV techniques. This was so weird, since no UVA relative content were introduced in previous paragraphs. Even, the authors did not know why they needed to use UAV platform for chlorophyll estimation.

2. in 2.2.1 part, for hyperspectral data acquisition, the authors must explain details for reflectance calibration to verify the feasibility to use the acquired reflectance for chlorophyll retrieval. this is the basis for the whole retrieval process. It is recommended to present some reflectance that acquired to see the difference between varied growth stages.

3. in section 2.2.2, for SPAD values, these values were just relative chlorophyll readings, cannot represent chlorophyll content. Only if these values were calculated with some conversion equations. Moreover, converted SPAD values could merely represent leaf chlorophyl content. For canopy chlorophyll content calculation, LAI values were still needed. So, the conception CCC in the whole manuscript were not correct.

4. in section 3, for results presentation. The authors should present some scatterplots of measured SPAD values and predicted values, since RMSE, RPD indictors were not enough for comparing the results estimated with different methods for different growth periods.

Specific comments

1. for different nonlinear fitting methods in 2.4.2, such as PLSR, SVM, BPNN, key parameters that used for these methods should be clearly mentioned.

2. Grammer of the manuscript should be checked throughout the whole manuscript.

Overall English is fine, Minor editing of English language is required throughout the whole manuscript.

Reviewer 3 Report

Dear Authors,

The subject of the study is interesting and topical, with scientific and practical importance.

The introduction is presented correctly, in accordance with the subject.

Numerous bibliographic sources were consulted, but were not presented in the References chapter.

Methodology of the study was clearly presented, and appropriate to the proposed objectives.

The obtained results are important and have been analyzed and interpreted correctly, in accordance with the current methodology.

The discussions are appropriate, in the context of the results, and was conducted compared to other studies in the field.

The scientific literature could not be evaluated, because the entire content of the References chapter is missing from the article.

Some suggestions and corrections were made in the article.

The following aspects are brought to the attention of the authors.

1.

Page 6, row 186

“Equation (1)” instead of “(1)”

2.

Page 13, row 443

“R2” instead of “R2”

3.

References

The entire References chapter is missing from the article.

Please, when entering the References content, check that all bibliographic sources cited in the text are included in the References chapter, and vice versa.

Also, check that the settings correspond to the Instructions for Authors, and Microsoft Word template, Remote Sensing journal.

Round 2

Reviewer 2 Report

Thanks to the authors for considering the comments and suggestions. In the present version of the manuscript and cover letter, the authors have explained and answered general comments accordingly. Nevertheless, key problem, and novelty of the manuscript are still not mentioned clearly in the revised manuscript, and there are still two problems:

First one is the usage of canopy chlorophyll content (CCC). For this conception, it has a unit, but for SPAD readings, these values just represent relative chlorophyll content, thus these two things can not be replaceable. It is very necessary to change canopy chlorophyll content (or CCC) throughout the whole manuscript to some other expression that can stand for SPAD readings.

Next one is the added figure 7. I believe it must be a mistake since obvious error occurred in the figure. The range for the end of red band, red-edge wavebands, and NIR bands are wrong. This information must be corrected accordingly.

I believe these problems must be addressed before manuscript acceptance.

Grammar of the manuscript should be further improved throughout the whole manuscript.

Round 3

Reviewer 1 Report

Thanks for the authors' efforts to revise the paper. I am still not satisfied about the utilization of the random forest algorithm. I hope you test it in your study, rather than just change the introduction. But I understand, it might change the whole structure of the paper, which takes time. This is the reason why I suggested you to apply longer review time from the Journal. I would strongly suggest the authors to test this method in your future study. It is not hard at all; you can easily run it in any programming language, e.g., R, Python, Matlab and etc.